# Longitudinal Association of Universal Screening and Treatment for Major Depressive Disorder with Survival in Cancer Patients

**DOI:** 10.3390/jpm12081213

**Published:** 2022-07-26

**Authors:** Yung-Chieh Yen, Chin-Yu Huang, Hsue-Wei Chan, You-Yu Wang, Te-Chang Changchien, Deng-Wu Wang, Po-Chun Lin, Ting-Ting Chang, Yu-Wen Chiu

**Affiliations:** 1Department of Psychiatry, E-Da Hospital, Kaohsiung 824, Taiwan; jackycyen@yahoo.com (C.-Y.H.); ed107263@edah.org.tw (H.-W.C.); ed106579@edah.org.tw (Y.-Y.W.); ed106351@edah.org.tw (T.-C.C.); ed109635@edah.org.tw (D.-W.W.); ed108897@edah.org.tw (P.-C.L.); ed104871@edah.org.tw (T.-T.C.); ed113948@edah.oeg.tw (Y.-W.C.); 2School of Medicine, I-Shou University, Kaohsiung 824, Taiwan

**Keywords:** cancer, major depressive disorder, survival

## Abstract

Evidence for clinical screening and intervention for depression in cancer and the effect of this intervention on cancer prognosis is suboptimal. This study substantialized a complete model with universal screening and intervention for major depressive disorder (MDD) and explored its effect on survival in patients. This longitudinal study recruited cancer patients routinely screened for MDD with a two-stage model. Data including sex, age, cancer diagnosis, first diagnosis date, date of death, cancer stage, and MDD diagnosis and treatment were collected from medical records and the national registration system for cancer. Kaplan–Meier’s survival analysis and the Cox proportional hazards regression model were applied to analyze the effects of associated factors on survival. Further subgroup analysis for 14 types of cancer primary site was also performed. Overall, the hazard for patients adhering to psychiatric treatment for MDD before cancer diagnosis was not statistically different from that for patients without MDD (hazard ratio (HR) = 1.061, 95% CI: 0.889–1.267, *p* = 0.512). The hazard for patients adhering to psychiatric treatment after cancer diagnosis was significantly lower than that for patients without MDD (HR = 0.702, 95% CI: 0.607–0.812, *p* < 0.001). Those who were diagnosed with MDD after cancer diagnosis and adhered poorly to psychiatric treatment had the greatest hazard (HR = 1.829, 95% CI: 1.687–1.984, *p* < 0.001). The effect of intervention for MDD varied across different primary cancer types.

## 1. Introduction

Among all diseases, cancer poses the highest clinical, social, and economic burden in terms of cause-specific disability-adjusted life years globally [1]. In 2020, 19.3 million new cancer cases and almost 10.0 million cancer deaths were estimated. Preceding lung, colorectal, prostate, and stomach cancers, breast cancer in women is the most diagnosed cancer, with an estimated 2.3 million new cases. Lung cancer remains the leading mortality cause, with an estimated 1.8 million deaths, followed by colorectal, liver, and stomach cancers and breast cancer in women. Global cancer cases are expected to be 28.4 million in 2040, a 47% increase from the 2020s [2]. Cancer is the leading mortality cause globally, followed by ischemic heart disease, and it will still be the leading cause of death in 2060 (~18.63 million deaths) [1]. Over the last three decades, cancer mortality has declined overall in most high-income countries, except for mortality due to pancreatic and lung cancers in women. Moreover, cancer risk factors, screening tools, and diagnostic practices have changed with advances in treatment [3]. In Taiwan, cancers have been the leading mortality cause for 40 years. The four most common cancers are colorectal, lung, breast, and liver cancers; the top four leading causes of cancer death are lung, liver, colorectal, and breast cancers. While combatting cancers, new interventions in surgery, radiotherapy, chemotherapy, and target therapy as well as advanced prognostic factors such as biomarkers, nutrition state [4], and depression [5,6,7,8,9] all play critical roles. Compared to the focus on treatments for cancer itself, less focus has been provided to interventions for managing modifiable prognostic factors. The prevalence of depression and depressive disorders in cancer patients is between 5% and 60% [10]. Patients with depression experience helplessness, hopelessness, and negative emotions, which may discourage them from adhering to medical advice for cancer [11,12]. Stress or depression may induce cancer pathogenesis or progression as psychosocial stressors promote inflammation and oxidative/nitrosative stress, decreased immunosurveillance, and dysfunctional activation of the autonomic nervous system and Hypothalamic-Pituitary-Adrenal (HPA) axis [13]. The molecular mechanism underlying chronic stress, involving the continuous release of neurotransmitters from the neuroendocrine system, also affects breast cancer occurrence and development [14]. Meanwhile, cancer-induced activation of the immune system, blood–brain barrier breakdown, and chronic neuroinflammation may result in patients’ depression and memory impairment [15]. Furthermore, a recent meta-analysis revealed that depression is significantly associated with higher breast cancer recurrence, all-cause mortality, and cancer-specific mortality [6]. In a 3-year follow-up study of patients with prostate cancer undergoing radical prostatectomy, depression was correlated with an unfavorable survival profile [7]. Similarly, in two other follow-up studies, depression was correlated with the worse survival prognosis of colorectal cancer [9,11]. In a large-scale follow-up study, major depressive disorder (MDD) was associated with worse survival in patients with several common cancers, with a similarly increased hazard value [16]. In two similar studies, both depression and inflammation independently predicted inferior survival in patients with advanced lung cancer. Inflammation-induced worsening of patient survival is further mediated by depression, implying the potential role of depression (a treatable prognostic factor) in cancer progression [8,17]. Considerable health-care resources have long been devoted to cancer care globally. However, appropriate assessment and management of depressive disorders in cancer patients remain insufficient [10]. Most cancer patients with MDD fail to receive effective treatment for depression [18]. Two recent guidelines have suggested screening, assessment, and appropriate management of depression among adult cancer patients [19,20]. These guidelines recommend universal screening with appropriate tools and further referral to psychiatric professionals when required. Psychiatric treatment provided by a multidisciplinary team that incorporates wholistic care for depression is required. However, suboptimal evidence is available for extensive clinical screening and an intervention program for depression in cancer, as well as for the effect of the intervention on cancer prognosis. Several meta-analyses have concluded that psychosocial interventions for cancer, especially nonmetastatic cancers, may offer short-term survival benefits [21,22,23]; however, these studies have failed to provide evidence related to the benefits of interventions for cancer patients with MDD. In a randomized control trial (RCT), a specific MDD treatment program was effective in improving depression and quality of life in cancer patients; however, a significant effect on survival was not observed [24]. Therefore, our study substantialized a complete model with universal screening and intervention for MDD and explored its effectiveness in improving the survival of patients with different cancer types in real-world large-scale clinical practice.

## 2. Materials and Methods

### 2.1. Design and Setting

In this longitudinal study involving survival analysis, the medical records of adult cancer patients in the E-Da Hospital from October 2009 to September 2020 were retrieved. The E-Da Hospital is a non-profit academic medical center in southern Taiwan with more than 1200 beds and all specialties of clinical medicine, including oncology and psychiatry. The hospital’s IRB approved the research. Since 2009, the hospital has been routinely screening for depression among all cancer inpatients by using a two-stage model [25]. In the first stage, patients complete the Taiwanese Depression Questionnaire (TDQ) [26] after admission. If the TDQ score is 13 or more, patients are promptly referred to board-certified psychiatrists. In the second stage, these psychiatrists confirm MDD diagnosis and arrange psychiatric treatment. MDD exclusion was based on psychiatric assessment or a TDQ score of 12 or less. Those excluded were rescreened for MDD by using the same process at the next admission arranged for cancer treatment. Cancer diagnosis and treatment in the hospital fulfill the criteria of the Cancer Control Act and the relevant regulations for cancer intervention in Taiwan. This standardized management is routinely provided to every cancer patient according to their diagnosis, cancer stage, and choice of treatment. Management of both cancers and MDD is covered by national health insurance.

### 2.2. MDD Intervention

In this study, comprehensive psychiatric treatment included, but was not limited to, ambulatory care, hospitalization, and psychiatric emergency care with pharmacotherapy and psychotherapy. Treatment planning involved biological, psychological, and social aspects as well as a stepped care intervention model based on the current service structure and patients’ preference [19]. The recent finding of inflammatory cytokine-associated depression being a specific subtype of depression with particular relevance in cancer opened new pathways to explore therapeutic targets and biobehavioral interventions [27]. For the analyses, all patients were categorized into the following clinical groups based on the presence or absence of MDD and their adherence to psychiatric treatment: (1) MDD diagnosed and treated before cancer diagnosis; (2) MDD diagnosed after cancer diagnosis with adherence to psychiatric treatment; (3) MDD diagnosed after cancer diagnosis with nonadherence to psychiatric treatment; and (4) no MDD (Figure 1). Adherence to psychiatric treatment for MDD implied that patients complied with psychiatrists’ medical advice along the course of follow-up for more than 6 months until they were recommended to discontinue psychiatric treatment; whereas non-adherence implied that patients failed to comply with continuing psychiatric treatment suggested by psychiatrists, although cancer treatment was continued.

### 2.3. Primary Cancer Type

Cancer diagnosis, first diagnosis date, and cancer stage were obtained from medical records and further confirmed by national registration databases. Primary cancer sites were categorized into 14 groups according to the ICD-O-3 topography codes. Cancer stage classification, except for lymphoma and leukemia, was based on the TNM staging system created and updated by the American Joint Committee on Cancer and the International Union Against Cancer. Lymphoma staging was based on Lugano classification, and leukemia was classified into four diagnoses: acute lymphoblastic leukemia, acute myeloid leukemia, chronic lymphoblastic leukemia, and chronic myeloid leukemia.

### 2.4. Mortality

The endpoint of the study was death, unless censored for this study. Information on the date and cause of death was collected and confirmed based on medical records and the registration system. All-cause mortality, including intentional self-harm, was recorded.

### 2.5. Statistical Methods

We censored the data of participants on 31 December 2020, and analyzed survival as a trial outcome. Through Kaplan–Meier’s survival analysis (log-rank test), we explored the relationship between psychiatric treatment for MDD and the survival of cancer patients, aiming to clarify the effect of the intervention on survival in the presence of MDD and from the perspective of adherence to psychiatric treatment. Using the Cox proportional hazards regression model, we analyzed the effects of associated factors (sex, age at cancer diagnosis, age at screening for MDD, cancer site and stage, and MDD diagnosis and treatment) on the survival of cancer patients. Further subgroup analysis of 14 types of cancer primary site was performed by using the same procedure in order to explore if effects differed between cancer types.

## 3. Results

### 3.1. Demographics

Overall, 10,534 cancer patients were recruited after a thorough review of hospital medical records and national registration data. Among them, most were male patients (67.0%) who were categorized into Group 4 (85.1%), with a mean age at cancer diagnosis of 61.1 years (SD: 13.0), mean age at screening for MDD of 61.9 years (SD: 12.9), and mean age at death of 65.0 years (SD: 13.0) (Table 1).

The overall mean survival period was 7.00 years (SD: 0.29) (Table 2). Patients with earlier stages of cancer tended to survive longer. The mean survival period was longer in Group 2. Based on screening using the TDQ and further confirmation by the psychiatrists, 1319 (12.5%) of the cancer patients required psychiatric treatment; of them, 882 (8.4%) were suggested to undergo, but failed to receive, regular psychiatric treatment, and 437 (4.1%) were diagnosed with MDD and adhered to regular psychiatric treatment as suggested. Moreover, 254 (2.4%) of the cancer patients had received psychiatric treatment before cancer diagnosis; their psychiatric treatment was continued with cancer treatment.

### 3.2. Survival Analyses

#### 3.2.1. Kaplan-Meier’s Analysis

Using Kaplan–Meier’s survival analysis (log-rank test), significant differences were observed among the four cancer stages (chi-square 1819.07, df = 3, *p* < 0.001), four groups of cancer patients according to their depressive disorders and adherence to psychiatric treatment (chi-square = 267.17, df = 3, *p* < 0.001), and 14 primary cancer types (chi-square = 1285.11, df = 13, *p* < 0.001) (Table 1 and Figure 2). A more favorable survival outcome was observed in patients who were in the earlier stages of cancer, who were having MDD after cancer diagnosis yet adhering to psychiatric treatment (i.e., Group 2), and who had one of several types of cancer (e.g., thyroid, head/neck, and breast cancers).

#### 3.2.2. Cox Proportional Hazards Model

In the Cox proportional hazards regression model incorporating sex, age at cancer diagnosis, age at screening for MDD, cancer stage, MDD intervention group, and primary cancer type, patients fulfilling the following criteria had higher survival chances: female sex, younger age at cancer diagnosis, older age at screening for MDD, earlier cancer stage, and adherence to psychiatric treatment for MDD after cancer diagnosis, that is, Group 2 (Table 3 and Figure 3). According to this model, male patients had a higher risk of death than female patients (hazard ratio HR = 1.179, 95% CI: 1.103–1.261, *p* < 0.001). Moreover, an older age at cancer diagnosis by 1 year was associated with a 21.2% increase in mortality risk (HR = 1.212, 95% CI: 1.193–1.231, *p* < 0.001), and an older age at screening for MDD by 1 year was associated with a 15.7% decrease in mortality risk (HR = 0.843, 95% CI: 0.830–0.856, *p* < 0.001). The hazard for patients adhering to psychiatric treatment for MDD before cancer diagnosis (Group 1) was not statistically different from that for patients without MDD (Group 4) (HR = 1.061, 95% CI: 0.889–1.267, *p* = 0.512). The hazard for patients adhering to psychiatric treatment for MDD after cancer diagnosis (Group 2) was significantly lower than that for Group 4 (HR = 0.702, 95% CI: 0.607–0.812, *p* < 0.001). By contrast, Group 3 (those diagnosed to have MDD after cancer diagnosis and adhering poorly to psychiatric treatment) had a greater hazard than Group 4 (HR = 1.829, 95% CI: 1.687–1.984, *p* < 0.001).

Furthermore, having cancer at specific sites, such as the liver/pancreas and esophagus, had the worst impact on survival after adjustment for sex, age at cancer diagnosis, age at screening for MDD, cancer stage, and MDD treatment group (HR = 1.630, 95% CI: 1.186–2.239, *p* = 0.003; HR = 1.482, 95% CI: 1.066–2.061, *p* = 0.019, respectively). By contrast, cancers at other sites, such as gynecological, genitourinary, head/neck, breast, and colorectal cancers, had the least impact on survival after adjustment for sex, age at cancer diagnosis, age at screening for MDD, cancer stage, and MDD treatment group (HR = 0.179, 95% CI: 0.112–0.286, *p* < 0.001; HR = 0.485, 95% CI: 0.349–0.673, *p* < 0.001; HR = 0.496, 95% CI: 0.361–0.681, *p* < 0.001; HR = 0.512, 95% CI: 0.360–0.728, *p* < 0.001; HR = 0.552, 95% CI: 0.400–0.763, *p* < 0.001, respectively).

### 3.3. Subgroup Analysis of 14 Primary Cancer Types

Appendix A shows the survival outcomes of the Cox proportional hazards regression model stratified by 14 primary cancer types after adjustment for sex, age at cancer diagnosis, age at screening for MDD, and cancer stage. Subgroup analysis of 14 primary cancer types revealed the heterogenous effect of MDD treatment on survival (Table 4). The comparison of Groups 1 and 4 revealed that only patients with skin cancer had a significantly higher survival hazard if they were diagnosed with MDD before cancer diagnosis (HR = 11.21, 95% CI: 2.215–56.72, *p* = 0.003); patients with the remaining 13 cancers did not have a significantly higher survival hazard if they were diagnosed with MDD and were treated for MDD before cancer diagnosis. In the comparison of Groups 2 and 4, only patients with liver/pancreatic, lung, esophagus, and hematological cancers had a significantly lower survival hazard if they were diagnosed with MDD and adhered to MDD treatment after cancer diagnosis (HR = 0.674, 95% CI: 0.459–0.990, *p* = 0.044; HR = 0.631, 95% CI: 0.437–0.911, *p* = 0.014; HR = 0.463, 95% CI: 0.235–0.912, *p* = 0.026; HR = 0.436, 95% CI: 0.212–0.895, *p* = 0.024, respectively), whereas patients with the remaining 10 primary cancer types did not exhibit a significantly higher survival hazard if they were diagnosed with MDD and adhered to MDD treatment after cancer diagnosis. On the other hand, only Group 3, with gastrointestinal, bladder, and skin cancers, exhibited no significant impact on survival compared with Group 4 (HR = 1.557, 95% CI: 0.997–2.430, *p* = 0.051; HR = 1.665, 95% CI: 0.778–3.564, *p* = 0.189; HR = 2.734, 95% CI: 0.655–11.41, *p* = 0.168, respectively), whereas Group 3 with the remaining 11 primary cancer types exhibited a significant negative impact on survival compared to Group 4. Sex was not a significant survival predictor in 11 primary cancer types. However, male patients with lung and thyroid cancers had shorter survival than female patients (HR = 1.565, 95% CI: 1.363–1.797, *p* < 0.001; HR = 2.346, 95% CI: 1.096–5.025, *p* = 0.028, respectively). Male patients with genitourinary cancer had longer survival than female patients (HR = 0.608, 95% CI: 0.473–0.781, *p* < 0.001). Older age at cancer diagnosis had an overall significant negative impact on survival in all 14 primary cancer types. Older age at screening for MDD had a significantly positive impact on survival in 12 of the 14 primary cancer types, except for patients with gynecological and thyroid cancers (HR = 0.913, 95% CI: 0.833–1.001, *p* = 0.052; HR = 0.892, 95% CI: 0.753–1.058, *p* = 0.190, respectively).

## 4. Discussion

Identification of MDD and its appropriate treatment, especially after cancer diagnosis, are associated with longer or equal survival among patients with all cancer types, except for skin cancer. Cancer patients with MDD and poor adherence to MDD treatment have the lowest probability of survival; this was observed in patients with all cancer types, except for those with gastrointestinal, bladder, and skin cancers. Aggressive universal screening and treatment for psychiatric disorders, such as MDD, in all cancer patients are promising for improving survival. Our study outcome is consistent with the findings from a previous meta-analysis of 15 RCTs on psychosocial interventions for cancer patients; according to that study, adequate intervention may reduce mortality risk with a risk ratio of 0.69 (95% CI: 0.55–0.87) in the first 2 years [21], although this meta-analysis did not focus on cancer patients with MDD. Our finding is also consistent with that of another meta-analysis of 13 RCTs on psychosocial interventions for cancer patients [22]. In this meta-analysis, a significant survival benefit was noted for the intervention group at 1 year (RR = 0.82, 95% CI: 0.67–1.00) and 2 years (RR = 0.86, 95% CI: 0.78–0.95). However, another meta-analysis of 15 RCTs concluded that psychosocial intervention is not associated with better overall survival (HR = 0.83, 95% CI: 0.68–1.10); only patients with nonmetastatic cancer at intervention implementation and a longer follow-up of >10 years exhibited a significant reduction in mortality (HR = 0.59, 95% CI: 0.49–0.71) [23]. The aforementioned three studies are not related to cancer patients with comorbid MDD. The only previous paper reporting survival outcomes from RCTs of interventions for comorbid MDD in cancer patients found no significant evidence confirming that it improved survival [24]. When it comes to observational studies, our finding that depression was associated with worse survival prognosis is consistent with several studies of patients with prostate cancer [7], colorectal cancer [9,11], and several common cancers [16]. However, our finding that screening and treatment for MDD was correlated to favorable survival prognosis of cancer has never been found before in any observational study.

Our study revealed overall beneficial effects (i.e., Group 2 versus Group 4) on survival among cancer patients with MDD because patients with depressive symptoms are likely to experience helplessness, hopelessness, and negative emotions, which may interfere with their motivation to adhere to medical treatment for cancer [11,12]. Treatment for MDD before cancer diagnosis may also prevent patients’ nonadherence to medical advice for cancer. Another possible explanation is that interventions for depressive symptoms are likely to improve cancer prognosis. Depression and cancer have reciprocal relationships in terms of occurrence, progression, and prognosis. Common underlying etiologies, for cancer and depression, such as HPA axis hyperfunction, are also underlying mechanisms for depression in cancer patients [28]. Cancer and anticancer treatments result in proinflammatory cytokine-mediated inflammation, thereby dysregulating HPA axis activity, which may lead to depression-like behavior. Depression may also cause HPA axis activation, which results in the downregulation of endogenous glucocorticoids, thereby leading to depressive signs and symptoms in cancer patients. All the aforementioned mechanisms indicate the potential role of MDD treatment and the necessity of developing new drugs or psychosocial interventions for cancer patients with MDD. All cancer patients must be evaluated for symptoms of depression periodically across the trajectory of care and treated according to the severity of their symptoms [20].

In our study, cancer patients were screened for depression whenever they were hospitalized for treatment, unless the previous evaluation of depression was within 1 month. Consequent MDD treatment was arranged immediately when required. This allowed us to understand the real-world view in caring for cancer patients; however, one associated limitation was the lack of detailed socio-economic data. Our study compared survival among patients with different primary cancer types in the same clinical setting. The subgroup analysis of 14 primary cancer types revealed that patients with 4 primary cancer types (liver/pancreatic, lung, esophagus, and hematological cancers) may benefit from MDD treatment and have longer survival, whereas patients with the remaining 10 types of other cancers may not. The overall probability of survival in patients with MDD receiving treatment before cancer diagnosis (i.e., Group 1 versus Group 4) is similar to that in cancer patients without MDD. However, one exception was noted in the subgroup analysis. Patients with skin cancer and premorbid MDD had shorter survival, even if they had received MDD treatment. On the other hand, patients with MDD who failed to receive treatment regularly (i.e., Group 3 versus Group 4) had the lowest probability of survival among those with 11 primary cancer types. Patients with gastrointestinal, bladder, and skin cancers who were diagnosed with MDD after cancer diagnosis, but were not treated for MDD, did not have significantly shorter survival. Consequently, MDD and its interventions may play different roles in different primary cancer types, and this observation should be considered in further research aiming to clarify the effects of MDD and MDD treatment on the survival of cancer patients. In our study, 14.9% of all cancer patients had MDD. The prevalence of depression in cancer patients is high and heterogenous among different cancer types, cancer stages, assessment timing, and tools or measures of evaluation [29]. Among all cancer patients with MDD in our study, 56.4% failed to adhere to MDD treatment. The proportion of patients undergoing effective treatment for MDD was not satisfactory as it was in a previous study [18]. Thus, the guideline for screening of MDD and adherence to MDD treatment in cancer patients is not widely accepted.

## 5. Conclusions

Universal screening and treatment for MDD after cancer diagnosis is associated with an improvement in cancer patients’ survival of 29.8% on average, especially among those with liver/pancreatic, lung, esophagus, and hematological cancers. Cancer patients diagnosed as having MDD (after screening) but not adhering to MDD treatment have higher chances of death by 82.9%, except for patients with gastrointestinal, bladder, and skin cancers. The chances of survival for cancer patients with premorbid MDD who are undergoing continuing treatment for MDD are similar to those of patients without MDD, except among patients with skin cancer.

## Figures and Tables

**Figure 1 jpm-12-01213-f001:**
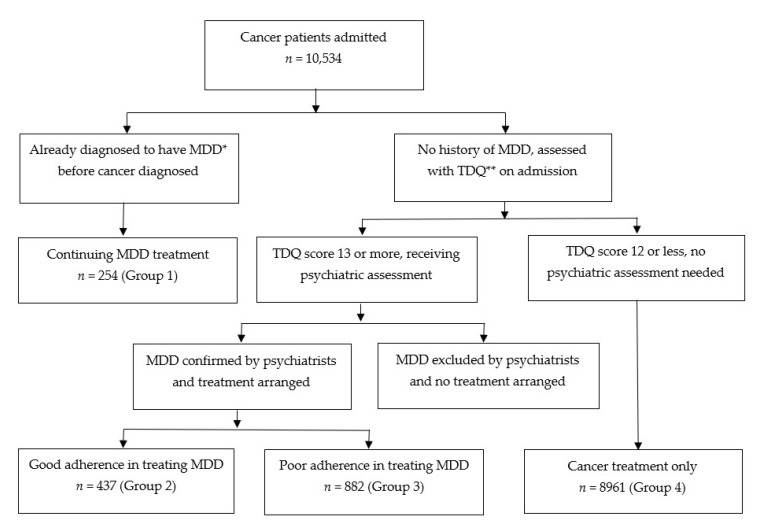
The flow of screening, intervention for MDD, and stratification of study groups. * MDD: major depressive disorder; ** TDQ: Taiwanese Depression Questionnaire.

**Figure 2 jpm-12-01213-f002:**
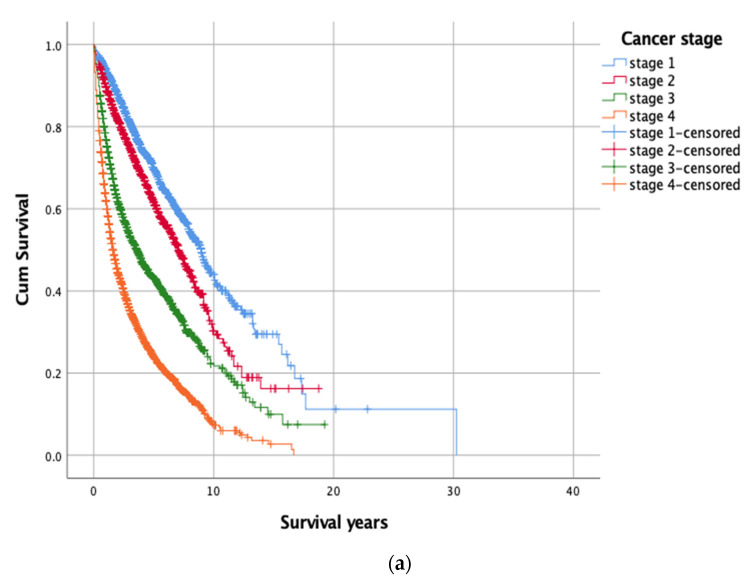
Kaplan-Meier’s survival functions stratified by (**a**) stage of cancer, and (**b**) primary cancer type.

**Figure 3 jpm-12-01213-f003:**
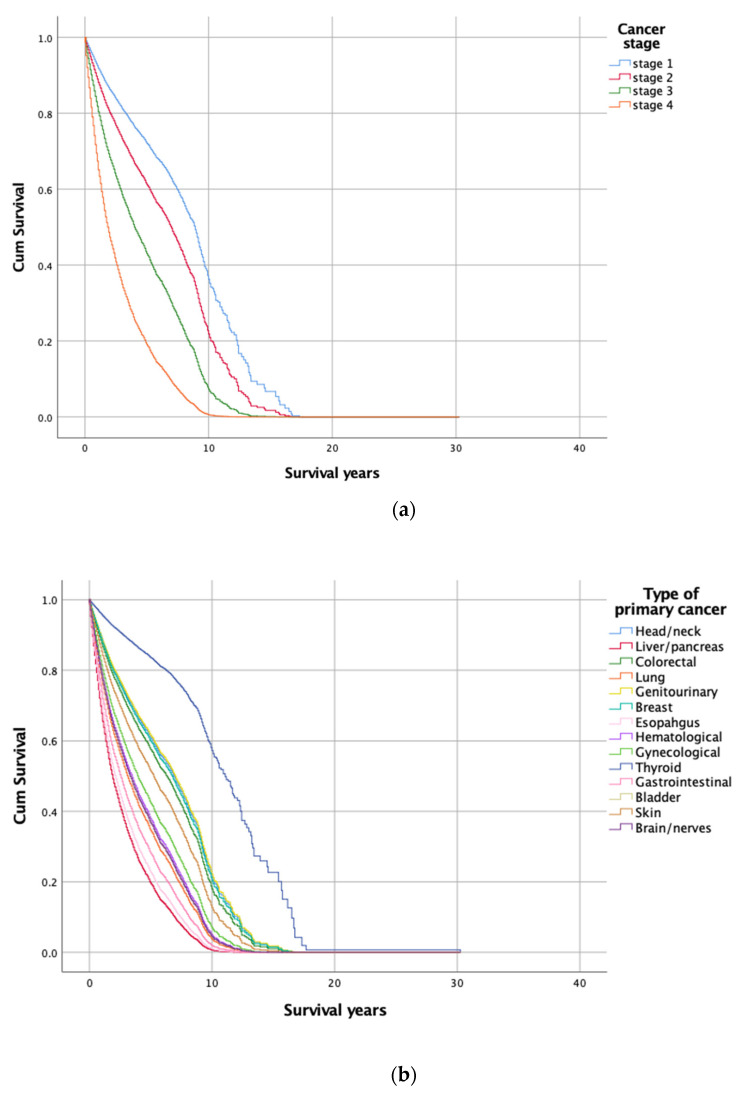
Survival functions of the Cox proportional hazard model stratified by (**a**) stage of cancer, adjusting for sex, age of cancer diagnosis, age of screening for MDD, primary cancer type, and MDD intervention group, and (**b**) primary cancer type, adjusting for sex, age of cancer diagnosis, age of screening for MDD, stage of cancer, and MDD intervention group.

**Table 1 jpm-12-01213-t001:** Characteristics of subjects (*n* = 10,534).

Characteristics	Number (%)	Mean (SD)
Male	7062 (67.0)	
Age at diagnosis of cancer (year)	10,534 (100)	61.1 (13.0)
Age at death (year)	5394 (51.2)	65.0 (13.0)
Age at screening MDD (year)	10,534 (100)	61.9 (12.9)

SD: standard deviation; MDD: major depressive disorder.

**Table 2 jpm-12-01213-t002:** Survival time by cancer stage, intervention group *, and primary cancer type (*n* = 10,534).

Survival Time (Year)	Number (%)	Censored (%)	Mean (SD)	*p ***
Overall	10,534 (100)	5140 (48.8)	7.00 (0.29)	
Cancer stage 1	2859 (27.1)	1981 (69.3)	10.84 (0.75)	<0.001
Cancer stage 2	1719 (16.3)	1088 (63.3)	8.01 (0.33)	
Cancer stage 3	2390 (22.7)	1104 (46.2)	5.86 (0.24)	
Cancer stage 4	3566 (33.9)	967 (27.1)	3.39 (0.10)	
MDD intervention group 1	254 (2.4)	122 (48.0)	7.67 (0.58)	<0.001
MDD intervention group 2	437 (4.1)	244 (55.8)	8.14 (0.40)	
MDD intervention group 3	882 (8.4)	148 (16.8)	4.16 (0.33)	
MDD intervention group 4	8961 (85.1)	4626 (51.6)	6.42 (0.13)	
Head/neck (C00–13, C30–32, C77)	2314 (22.0)	1312 (56.7)	9.85 (0.69)	<0.001
Liver/pancreas (C22–C35)	1780 (16.9)	596 (33.5)	4.73 (0.17)	
Colorectal (C18–21)	1365 (13.0)	787 (57.7)	6.59 (0.28)	
Lung (C33–38, C45)	1301 (12.4)	362 (27.8)	3.33 (0.17)	
Genitourinary (C60–68)	1085 (10.3)	677 (62.4)	6.94 (0.24)	
Breast (C50)	518 (4.9)	363 (70.1)	8.58 (0.50)	
Esophagus (C15)	498 (4.7)	133 (26.7)	3.16 (0.17)	
Hematological (C7A, C81–94)	373 (3.5)	155 (41.6)	5.19 (0.49)	
Gynecological (C51–58)	358 (3.4)	113 (31.6)	7.33 (0.79)	
Thyroid (C73)	337 (3.2)	305 (90.5)	12.4 (0.98)	
Gastrointestinal (C16–17)	290 (2.8)	100 (34.5)	3.96 (0.35)	
Bladder (C67)	152 (1.4)	54 (35.5)	6.50 (0.55)	
Skin (C43–44)	102 (1.0)	62 (60.8)	7.02 (1.00)	
Brain/nerves (C70–72, C79)	61 (0.5)	21 (34.4)	4.63 (0.85)	

* Groups: (1) MDD diagnosed and treated before the diagnosis of cancers; (2) MDD diagnosed after the diagnosis ancers with good adherence; (3) MDD diagnosed after the diagnosis of cancers with poor adherence; (4) no MDD. ** log-rank test of equality of survival distribution for the different levels. SD: standard deviation; MDD: major depressive disorder.

**Table 3 jpm-12-01213-t003:** The Cox proportional hazards regression model * predicting mortality of all cancer patients (*n* = 10,534).

Characteristics		HR	95% CI		*p*
Male (vs. female)		1.179	1.103	1.261	<0.001
Age at the diagnosis of cancer (per year)		1.212	1.193	1.231	<0.001
Age at screening of MDD (per year)		0.843	0.830	0.856	<0.001
Cancer stage	1	0.197	0.181	0.214	<0.001
	2	0.295	0.270	0.323	<0.001
	3	0.508	0.474	0.546	<0.001
	4	reference	-	-	-
MDD intervention group **	1	1.061	0.889	1.267	0.512
	2	0.702	0.607	0.812	<0.001
	3	1.829	1.687	1.984	<0.001
	4	reference	-	-	-
Head/neck		0.496	0.361	0.681	<0.001
Liver/pancreas		1.630	1.186	2.239	0.003
Colorectal		0.552	0.400	0.763	<0.001
Lung		1.061	0.772	1.458	0.714
Genitourinary		0.485	0.349	0.673	<0.001
Breast		0.512	0.360	0.728	<0.001
Esophagus		1.482	1.066	2.061	0.019
Hematological		0.970	0.692	1.361	0.861
Thyroid		0.858	0.602	1.223	0.396
Gynecological		0.179	0.112	0.286	<0.001
Gastrointestinal		1.268	0.900	1.786	0.175
Bladder		1.002	0.691	1.453	0.991
Skin		0.664	0.427	1.033	0.069
Brain/nerves		reference	-	-	-

* In this model, sex, age at the diagnosis of cancer, age at screening of MDD, cancer stage, MDD intervention group, and primary type of cancer were incorporated as independent variables. ** Groups: (1) MDD diagnosed and treated before the diagnosis of cancers; (2) MDD diagnosed after the diagnosis of cancers with good adherence; (3) MDD diagnosed after the diagnosis of cancers with poor adherence; (4) no MDD. HR: hazard ratio; CI: confidence interval; MDD: major depressive disorder.

**Table 4 jpm-12-01213-t004:** Cox proportional hazards regression model * predicting mortality by primary cancer type.

**Head/Neck**		**HR**	**95% CI**		** *p* **	**Hematological**		**HR**	**95% CI**		* **p** *
Male (vs. female)		1.092	0.873	1.092	0.443	Male (vs. female)		0.957	0.729	1.255	0.748
Age **		1.114	1.080	1.149	<0.001	Age		1.307	1.184	1.443	<0.001
Screening age ***		0.916	0.889	0.945	<0.001	Screening age		0.798	0.724	0.881	<0.001
Group ****	1	1.008	0.640	1.588	0.971	Group	1	0.943	0.437	2.038	0.882
	2	0.806	0.597	1.090	0.162		2	0.436	0.212	0.895	0.024
	3	1.976	1.687	2.314	<0.001		3	1.928	1.319	2.819	0.001
	4	reference	-	-	-		4	reference	-	-	-
**Liver/Pancreas**		**HR**	**95% CI**		** *p* **	**Gynecological**		**HR**	**95% CI**		** *p* **
Male (vs. female)		1.078	0.948	1.226	0.254	Male (vs. female)		-	-		-
Age		1.327	1.282	1.374	<0.001	Age		1.114	1.014	1.223	0.024
Screening age		0.764	0.738	0.791	<0.001	Screening age		0.913	0.833	1.001	0.052
Group	1	0.889	0.610	1.296	0.540	Group	1	0.208	0.024	1.771	0.151
	2	0.674	0.459	0.990	0.044		2	0.838	0.303	2.320	0.734
	3	1.448	1.176	1.783	<0.001		3	2.835	1.745	4.605	<0.001
	4	reference	-	-	-		4	reference	-	-	-
**Colorectal**		**HR**	**95% CI**		** *p* **	**Thyroid**		**HR**	**95% CI**		** *p* **
Male (vs. female)		1.085	0.913	1.289	0.356	Male (vs. female)		2.346	1.096	5.025	0.028
Age		1.149	1.095	1.206	<0.001	Age		1.207	1.015	1.436	0.034
Screening age		0.893	0.851	0.937	<0.001	Screening age		0.892	0.753	1.058	0.190
Group	1	1.008	0.610	1.664	0.976	Group	1	2.721	0.517	14.33	0.238
	2	0.704	0.449	1.105	0.127		2	0.539	0.070	4.139	0.552
	3	1.646	1.256	2.157	<0.001		3	12.142	3.279	44.96	<0.001
	4	reference	-	-	-		4	reference	-	-	-
**Lung**		**HR**	**95% CI**		** *p* **	**Gastrointestinal**		**HR**	**95% CI**		** *p* **
Male (vs. female)		1.565	1.363	1.797	<0.001	Male (vs. female)		1.071	0.791	1.451	0.656
Age		1.377	1.300	1.459	<0.001	Age		1.166	1.051	1.294	0.004
Screening age		0.739	0.698	0.783	<0.001	Screening age		0.877	0.791	0.973	0.013
Group	1	0.823	0.492	1.377	0.458	Group	1	0.582	0.140	2.426	0.457
	2	0.631	0.437	0.911	0.014		2	0.559	0.197	1.587	0.275
	3	1.662	1.361	2.031	<0.001		3	1.557	0.997	2.430	0.051
	4	reference	-	-	-		4	reference	-	-	-
**Genitourinary**		**HR**	**95% CI**		** *p* **	**Bladder**		**HR**	**95% CI**		** *p* **
Male (vs. female)		0.608	0.473	0.781	<0.001	Male (vs. female)		0.916	0.583	1.439	0.705
Age		1.201	1.134	1.273	<0.001	Age		1.332	1.191	1.490	<0.001
Screening age		0.861	0.813	0.912	<0.001	Screening age		0.774	0.693	0.865	<0.001
Group	1	1.265	0.706	2.265	0.429	Group	1	1.943	0.247	15.29	0.528
	2	0.719	0.473	1.093	0.123		2	1.285	0.509	3.243	0.595
	3	2.946	2.072	4.188	<0.001		3	1.665	0.778	3.564	0.189
	4	reference	-	-	-		4	reference	-	-	-
**Breast**		**HR**	**95% CI**		** *p* **	**Skin**		**HR**	**95% CI**		** *p* **
Male (vs. female)		<0.001	<0.001	<0.001	0.969	Male (vs. female)		1.225	0.623	2.447	0.566
Age		1.121	1.043	1.204	0.002	Age		1.446	1.102	1.897	0.008
Screening age		0.915	0.851	0.983	0.015	Screening age		0.707	0.538	0.929	0.013
Group	1	1.276	0.620	2.627	0.509	Group	1	11.21	2.215	56.72	0.003
	2	1.334	0.688	2.584	0.394		2	0.756	0.098	5.838	0.789
	3	2.208	1.353	3.601	0.002		3	2.734	0.655	11.41	0.168
	4	reference	-	-	-		4	reference	-	-	-
**Esophagus**		**HR**	**95% CI**		** *p* **	**Brain/Nerves**		**HR**	**95% CI**		** *p* **
Male (vs. female)		1.461	0.847	2.519	0.172	Male (vs. female)		1.517	0.682	3.373	0.307
Age		1.243	1.138	1.357	<0.001	Age		2.023	1.432	2.858	<0.001
Screening age		0.810	0.741	0.885	<0.001	Screening age		0.502	0.356	0.708	<0.001
Group	1	1.073	0.546	2.110	0.838	Group	1	2.346	0.700	7.861	0.167
	2	0.463	0.235	0.912	0.026		2	0.260	0.048	1.397	0.116
	3	1.479	1.119	1.955	0.006		3	3.292	1.033	10.49	0.044
	4	reference	-	-	-		4	reference	-	-	-

Note: Stages of cancers were included in the model but not shown here. * In this model, sex, age at the diagnosis of cancer, age at screening of MDD, cancer stage, MDD intervention group, and primary type of cancer were incorporated as independent variables. ** Age at the diagnosis of cancer (per year). *** Age at the screening for MDD (per year). **** Groups: (1) MDD diagnosed and treated before the diagnosis of cancers; (2) MDD diagnosed after the diagnosis of cancers with good adherence; (3) MDD diagnosed after the diagnosis of cancers with poor adherence; (4) no MDD. HR: hazard ratio; CI: confidence interval; MDD: major depressive disorder.

## Data Availability

Data are open to the public on request after proper review.

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
