# Peer review of "Longitudinal Association of Universal Screening and Treatment for Major Depressive Disorder with Survival in Cancer Patients"

_jpm, 2022, doi:10.3390/jpm12081213_

Round 1

Reviewer 1 Report

I think the study design is rather observational than interventional. The authors didn't control the MDD treatment and screening, and it is a retrospective study, so the lack of randomization seems inappropriate as a limitation.

What was the calculated sample size if the authors claim too small? 

Figure 1. The definition of asterisk.

Table 1. It is very unclear what the numbers mean, e.g. 'age at diagnosis of cancer' - does it mean how many people responded to this question; 'mean (SD)' - meaning of what?

How were the confounding factors in the regression model chosen? (value of B/beta?)

What is the raw data?

Please write in the footnote of Table 2 what the confounders are.

Why did the authors write lines 229-233?

Table 3 is unclear.

The authors should extend the discussion of the results by comparing them with similar studies (observational study design).

The conclusions are too firm based on the study design and limitations.

Author Response

Thanks for the comments and suggestions.

Indeed, this is a retrospective cohort study. The lack of randomization should not be a limitation for this study design. It is revised.

  1. What was the calculated sample size if the authors claim too small? 

The limitation of small sample size in the only RCT effective in improving depression but not significantly improving survival in cancer patients (reference 22 originally) was raised by the authors themselves. However, these lines related to small sample size have been omitted due to revision this time. We may not be able to calculate the exact needed sample size in our cohort study based on an RCT.

  1. Figure 1. The definition of asterisk.

The definitions are added.

  1. Table 1. It is very unclear what the numbers mean, eg. 'age at diagnosis of cancer' - does it mean how many people responded to this question; 'mean (SD)' - meaning of what?

Age at diagnosis of cancer means the age when the diagnosis of cancer was confirmed according to medical records and national cancer registration system. It is not necessarily the same age as they received the routine screening for MDD.

  1. How were the confounding factors in the regression model chosen? (value of B/beta?)

All independent variables were separated into three blocks. Block 1 included sex, age at cancer diagnosis, and age at MDD screening. Block 2 included cancer stage and MDD intervention. Block 3 included the primary type of cancer. Enter method was applied in Cox regression. Omnibus tests of model coefficients were applied after each step to make sure the addition of new variables improved the model fit. Values of B/beta and Wald were produced but not included in the tables.

  1. What is the raw data?

The raw data of this study are from the medical records (including the outcomes of routine depression screening for cancer patients and treatment of MDD) and the national dataset of cancer registration system (including cancer diagnosis, stage, and mortality).  

  1. Please write in the footnote of Table 2 what the confounders are.

It is added in the footnote of Table 2.

  1. Why did the authors write lines 229-233?

It is revised.

  1. Table 3 is unclear.

Appropriate footnote is added.

  1. The authors should extend the discussion of the results by comparing them with similar studies (observational study design).

Comparison with 4 observational studies are added in the Discussion.

  1. The conclusions are too firm based on the study design and limitations.

The conclusion is revised.

Reviewer 2 Report

1. Introduction - a bit long and need trimming

2. Materials -

2.1 please elaborate on the hospital setting

2.2 how was compliance measured? how do you consider non adherence? how many missing appointment?

2.4 regarding mortality did you exclude non oncology causes of death

3. Results

162-167 & 220-222 may not need to include all theses details

230-233 need to be incorporated in the graph/legend

4. Discussion

299-308 need not to be here may include a brief in the introduction

321-328 need to be rephrased and included in the methods

Author Response

Thanks for the comments and suggestions.

  1. Introduction - a bit long and need trimming 

It is revised.

  1. Materials - 

2.1 please elaborate on the hospital setting 

It is added.

2.2 how was compliance measured? how do you consider non-adherence? how many missing appointments?

The adherence to treatment for MDD was defined as receiving treatment for at least 6 months and continuing further treatment unless suggested to stop the treatment by the psychiatrists. Non-adherence to treatment for MDD was defined as receiving treatment less than 6 months or discontinuing treatment after 6 months without the recommendation of the psychiatrists. It may not be easy to measure non-adherence with missing appointments alone because those cancer patients in Taiwan might have to re-arrange their appointments for their various cancer therapies with the help of case managers, and sometimes they were hospitalized in cancer ward or psychiatric ward. The definition of adherence is briefly described and added to 2.2.

2.4 regarding mortality did you exclude non oncology causes of death 

In this study, all-cause mortality was applied.  

  1. Results 

162-167 & 220-222 may not need to include all these details 

It is revised and simplified.

230-233 need to be incorporated in the graph/legend

It is revised.

  1. Discussion 

299-308 need not to be here may include a brief in the introduction

These lines are summarized and moved to Introduction.

321-328 need to be rephrased and included in the methods

They are revised and shifted to Methods.

Round 2

Reviewer 1 Report

Table 1 is still unclear. Probably the authors wanted to provide the numbers of people from whom they received data on specific variables, but the current record is not legible. Additionally, it may be worth considering the division into two tables because the difference in the number of columns in the upper and lower part makes it difficult to read.

I propose a vertical arrangement of figures in Figure 2. It is currently illegible, similarly to figure 3. Please consider putting figure 4 in the supplements and increase the size.

Author Response

Thank you again for the comments and suggestions.

  1. Table 1 is still unclear. Probably the authors wanted to provide the numbers of people from whom they received data on specific variables, but the current record is not legible. Additionally, it may be worth considering the division into two tables because the difference in the number of columns in the upper and lower part makes it difficult to read.

It is a very good idea to divide Table 1 into two tables. It’s revised and more details are added separately.

  1. I propose a vertical arrangement of figures in Figure 2. It is currently illegible, similarly to figure 3. Please consider putting figure 4 in the supplements and increase the size.

The Figures 2 and 3 have been re-arranged. The Figure 4 is shifted to the Supplements, marked as Figure 3S, and increased in size.